# Developmental Dysplasia of the Hip: How Does Social Media Influence Patients and Caregivers Seeking Information?

**DOI:** 10.3390/children8100869

**Published:** 2021-09-29

**Authors:** Ashok Para, Brian Batko, Joseph Ippolito, Gabriel Hanna, Folorunsho Edobor-Osula

**Affiliations:** Department of Orthopaedics, Rutgers New Jersey Medical School, Newark, NJ 07103, USA; bdb103@njms.rutgers.edu (B.B.); ippolija@njms.rutgers.edu (J.I.); gh221@njms.rutgers.edu (G.H.); edoborof@njms.rutgers.edu (F.E.-O.)

**Keywords:** developmental dysplasia of the hip, social media, thematic analysis, quantitative analysis

## Abstract

Developmental dysplasia of the hip (DDH) is a common orthopaedic condition affecting newborns. The rapid and vast adoption of social media has changed how we access medical information. The aim of this study was to deepen the understanding of the impact of social media as a tool used by caregivers. A search was performed on the Facebook (FB), Twitter (TW), and YouTube (YT) platforms. Information was quantitatively assessed by category, and number of posts and users. Comments and posts from the social medial platforms were then qualitatively assessed by using a thematic analysis. 16 Facebook pages and groups, 135 YouTube videos, and 5 Twitter accounts related to DDH were identified across 15 countries. A total of 25,471 comments/tweets were recorded. Across the social media platforms, the most common comments theme was “information sharing” (36.1%). Facebook groups had a significantly greater number of comments that were characterized as “social media as a second opinion” in comparison to YouTube videos (*p* < 0.001), whereas YouTube videos had significantly fewer comments characterized as “sharing information” in comparison to Facebook groups and Facebook pages (*p* < 0.0001). Orthopaedic surgeons may utilize caregiver presence on social media as an opportunity to help share accurate information and facilitate informed decision-making.

## 1. Introduction

Developmental dysplasia of the hip (DDH) is a common orthopaedic condition affecting newborns, with a worldwide incidence of 11.5 cases per 1000 births [1]. DDH refers to a continuum of abnormalities in the immature hip that can range from subtle dysplasia to dislocation, making it an important cause of childhood disability [2]. Bracing has historically been considered first-line treatment in children younger than six months; however, more recent literature has revealed advantages in proceeding directly to surgery in patients with severe DDH and in patients with congenitally dislocated hips [2,3]. In patients with milder DDH who are initially managed with bracing, surgery is considered when bracing fails [2]. Parents and caregivers of children with DDH experience major struggles including emotional distress when learning of the diagnosis, adjusting to prolonged periods of bracing, and stress associated with the potential for surgery.

In 2019, it was estimated that 9 out of 10 adults in the USA use the internet [4]. The rapid and vast adoption of social media has changed how we access medical information. With an estimated 3.6 billion users around the world, social media platforms are more accessible than ever before and allow users to share content and communicate with one another on opposite sides of the world [5]. As of January 2021, Facebook, YouTube and Twitter had 2.7 billion, 2.3 billion and 353 million active users, respectively [6]. 

The internet has become an avenue for families of children in need of orthopaedic consultation to seek health-related information. Studies have shown that parents and caregivers utilize social media to gather information regarding advice on specific diseases, sharing personal experiences, and gaining knowledge about specific treatment processes [7,8,9]. Although orthopaedic consultation will ultimately determine final advice and management, social media platforms such as FB, YT and TW can serve as important sources of information for families of children with DDH concerning the disease process and the various modalities of management. Currently, there is a paucity of literature available on the types of information available to patients and caregivers on social media platforms pertaining to DDH. The aim of this study was to deepen the understanding of the impact of social media as a tool used by caregivers by assessing information available on DDH on popular social media platforms.

## 2. Materials and Methods

### 2.1. Platform Search 

Institutional Review Board approval was not obtained or required for this study. Information collected was publicly available information and no personal health information was documented. FB pages, FB groups, YT and TW were searched using the keyword “Developmental Dysplasia of the Hip” and “DDH.” The decision was made to exclude Instagram as there were not a sufficient number of accounts or data points to perform statistical analysis and compare with data obtained from Facebook, YouTube, and Twitter. Google was excluded as it is not considered a social media platform, but rather a search engine. Facebook groups and pages with more than 500 members/likes were selected. YouTube videos with more than 500 likes were selected while excluding duplicated videos, non-English speaking videos and irrelevant content. All Twitter accounts were selected regardless of the number of followers. Facebook groups were split into public and private groups. Access was requested for all private groups. Data points that were collected included the number of members, followers, likes, comments or tweets, country of origin, access status, and year of platform creation. Social media platform membership minimum criteria were used as previously described [10]. Based on the collected information, each search result was sorted into one of five classifications: medical institution, news, nonprofit organization, promotional information, or story sharing. 

### 2.2. Thematic Analysis

In order to prevent selection bias, a qualitative analysis was performed on all comments, post and post responses on FB, YT and TW. Comments were given a category by two authors (BB and GH) independently. The senior author (FO) resolved any discrepancies. An axial coding system was used to create comment themes [11]. The comment themes included: sharing information and advice, appreciation and success stories, emotional support, social media as a second opinion, advertisements, challenges and difficulties, and inequities and access issues. YT comments were placed into an online word cloud generator to determine text frequency and relevance (6775 words) [12].

### 2.3. Statistical Analysis

Statistical analysis was conducted with IBM SPSS version 27. A non-parametric one-way ANOVA test was utilized to analyze thematic comment distribution between social media platforms. When making multiple comparisons, a multiple-comparison correction was used to adjust the *p*-value for multiple comparisons. The level of significance was set at *p* = 0.05.

## 3. Results

### 3.1. General Demographic Results

Overall, 156 social media platforms were identified between 2009 and 2019. A total of 25,471 comments were identified from accounts spanning 15 countries. The United States (68.4%), UK (10.5%) and Australia (6.8%) accounted for most accounts. Non-profit (64.3%) and medical institution (16.3%) category types comprised most accounts. 

### 3.2. Facebook Pages

A total of 10 Facebook pages were identified, with 44,221 likes (mean of 4422.1 (range: 506 to 11,738)), 2938 posts (mean of 293.8 (range: 49 to 260)), and 11,003 comments (mean of 1100.3 (range: 193 to 2952)). Of the 10 pages identified, 5 originated in Australia (50%), 2 originated in the UK (20%), and the USA, Canada, and Jordan each had 1 page (10%), respectively. The most Facebook pages related to DDH were created in 2015, with three pages created (Figure 1).

### 3.3. Facebook Groups

A total of six Facebook groups were identified, with 26,206 total members [mean of 4368 (range: 1576 to 9380)) and 12,727 comments (mean of 2121 (range: 875 to 3620)). Of the six groups identified, two groups originated in the USA (33.3%), two groups originated in the UK (33.3%), and two groups originated in Australia (33.3%). The most Facebook groups related to DDH were created in 2016, with three groups created (Figure 1). 

### 3.4. YouTube

A total of 135 YouTube videos were identified, with 6,035,643 total views (mean of 44,709 (range: 546 to 1,088,762)) and 623 total comments (mean of 4.6 (range: 0 to 61)). 85 YouTube videos (63%) originated in the USA. The most YouTube videos related to DDH were created in 2011, with 25 videos created (Figure 1).

### 3.5. Twitter

A total of five Twitter pages were identified, with 3478 total followers (mean of 695.6 (range: 40 to 2017)) and 1096 total tweets (mean of 223.6 (range: 1 to 776)). Two Twitter pages (40%) were created in the USA.

### 3.6. Sub-Thematic Analysis

One hundred sixty (160) accounts were identified and categorized into five main groups (Figure 2). 25,471 comments were classified into eight themes (Table 1). Thematic analysis of the 25,471 comments was conducted (Table 2). 623 YouTube comments were analyzed using a word cloud generator and the distribution of the most utilized words was represented (Figure 3 and Figure 4, respectively).

### 3.7. Thematic Comment Distribution

Pairwise comparison demonstrated a significant difference between social media platforms in terms of amount of comments associated with thematic distribution (*p* < 0.0001). Multiple comparison analysis showed that FBgroups and FB pages had significantly higher number of comments across all themes compared to YouTube (*p* < 0.0001). No significant difference was found in the distribution of comment themes between Twitter and other social media platforms across the majority of themes (Table 3).

## 4. Discussion

### 4.1. Quantitative Analysis

Social media platforms are important assets to help educate caregivers and guide management of patients, but may currently be underutilized by pediatric orthopaedists [13]. This study establishes that families and caregivers of children affected by DDH frequently and consistently use social media platforms such as Facebook, YouTube and Twitter to share information, with an average of 2547 comments shared per year. 

Our analysis revealed that YouTube had the majority of social media content related to DDH, accounting for 135 out of 160 (84.4%) total social media accounts. 85 of the YT videos (63%) were created by users from the USA. With 6,035,643 total views and only 623 total comments, there was an discrepancy between viewership and the interactive nature of YT videos. 62 out of 135 videos (45.9%) had zero comments recorded. Furthermore, YouTube videos are often shared on various additional social media platforms, which may elevate the total number of views without giving viewers the opportunity to comment on the video on the YouTube platform and interact with one another. “Appreciation and Success Stories,” “Sharing information and Advice,” and “Information Regarding Technique and Anatomy” were the three most common themes, however Facebook groups and pages still contained a significantly higher number of comments across all themes compared to YouTube (Table 3). Therefore, YouTube videos seem to be an effective modality for orthopaedists to share information regarding DDH, however they may not be the best platform for caregivers to interact with one another and share their experiences. 

Our study found that FB was the most interactive social media platform, accounting for 23,730 out of 25,449 (93.2%) of total social media comments; 11,003 comments were shared on Facebook pages and 12,727 comments were shared on Facebook groups. Only a total of 3 out of 16 (18.8%) total Facebook pages/groups were located in the USA. Given that Facebook pages and groups are located in various countries and are heavily used by caregivers of DDH, Facebook may offer a concrete starting point for information distribution by orthopaedists around the world. 

Twitter contributed a minimal amount of information in terms of social media accounts and comments, contributing merely five accounts (3.1%) and 1096 total comments (4.3%). Two of the five Twitter pages originated from the United States. The character limit of 140 characters per Tweet may limit both orthopaedic surgeons and caregivers from sharing detailed explanations and personal experiences. Additionally, our study found that there were no DDH-related Twitter accounts created after 2015, suggesting that Twitter may have fallen out of favor as a social media platform that can effectively provide caregivers information about DDH (Figure 1).

### 4.2. Thematic Analysis

Comment analysis demonstrated that social media platforms provide an informational role (Table 1). Comments looking for information accounted for 36.1% of the social media comments. The comments pertained to the experiences related to the day-to-day care for a child with DDH (Table 1). These experiences were communicated via words and images, indicating an informative use of social media. Furthermore, sub-thematic comment analysis demonstrated that information regarding DDH on social media platforms tended to largely provide secondary and emotional information, while medical information was only found in a small percentage (Table 2). 

The personal nature of the comments was dominant, often experiences that simplified caregiver support. This phenomenon can be elucidated by the powerful effect of discourse, which improves the recognition of the child’s needs [14]. “Emotional Support” was the third most popular theme overall with 11.35% of total comments across the selected social media platforms. In this category, caregivers shared their emotional distress when facing the diagnosis of DDH, as well as the challenges they experienced during DDH management (Table 1). As such, social media platforms can serve as an outlet for parents and caregivers of children with DDH to aid their emotional adjustment [15].

Upon review of YT comments, “Appreciation and Success Stories” was the dominant theme, accounting for 53.9% of all comments. YouTube viewership was high with 6,035,643 total views, however interactions between viewers were low with only 623 total comments. Word cloud analysis revealed that “Surgery” (37 comments) and “Pain” (21 comments) were among the top 10 most frequently commented words on YouTube (Figure 3 and Figure 4).

Furthermore, caregivers often utilized Facebook groups as a means of obtaining a second opinion (Table 3). Under the “social media as a second opinion” category, caregivers utilized Facebook groups to post medical questions as case descriptions or ultrasound results. Responses ranged from offering medical advice to referring caregivers to orthopaedic surgeons. In most cases, non-medical members led the discussions. Although information sharing is valuable, the medical advice shared is at risk of being misleading due to the absence of monitoring by experienced orthopaedic surgeons [7].

Another theme that emerged from our analysis is “Techniques and Anatomy”, where information exchange regarding DDH was predominately technical. The majority of these comments came from Facebook pages and groups that are dedicated to research and address various topics including the surgical correction of DDH. Although this may be a useful tool for doctors to inform parents and spark crucial conversations, it is important to note the possibility of doctors using these platforms for advertisement rather than education. These apprehensions highlight the importance of encouraging orthopaedic societies to increase their online presence to provide accurate, peer-reviewed information that is easily understandable [16].

This study had several limitations. The first limitation included the use of “Developmental Dysplasia of the Hip” and “DDH” as the two keywords to search for DDH-related content, excluding search results that may have differed in terminology. However, the authors are not familiar with alternative nomenclature or words that describe this condition and believe, to the best of our knowledge, that our search was all encompassing. Another limitation is that only English-based content was selected and analyzed, which prevented the analysis of social media platforms in different languages. Access could not be obtained in 4 out of 10 (40.0%) Facebook private groups, which limited this study from capturing different types of discussions taking place in private groups. However, we were still able to analyze 25,449 individual comments across six countries, which provided a large comment base. Word cloud analysis was performed on YT comments could have generated a distribution that does not represent all social media platforms. In addition, word cloud generators analyze word frequency without any consideration for the context in which they were used [17]. Furthermore, this study represents data from a specific point in timebased off the current use of social media by caregivers as of data collection completion date, and reported numbers are likely to to change due to the dynamic nature of social media platforms. 

This study provides an understanding of the use of social media platforms as important tools for information sharing by caregivers of children with DDH. These platforms represent an opportunity for orthopaedic societies to share information supported by peer-reviewed evidence that may lead to improved patient care and superior health outcomes. Surgeons and societies should be encouraged to increase their presence on social media platforms to aid and educate parents and caregivers of children with DDH.

## Figures and Tables

**Figure 1 children-08-00869-f001:**
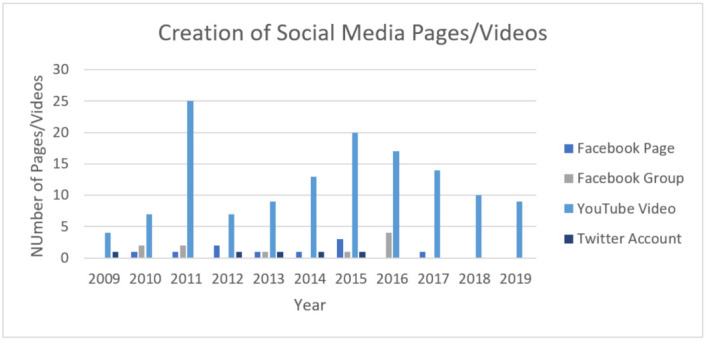
Social media platform accounts created by year.

**Figure 2 children-08-00869-f002:**
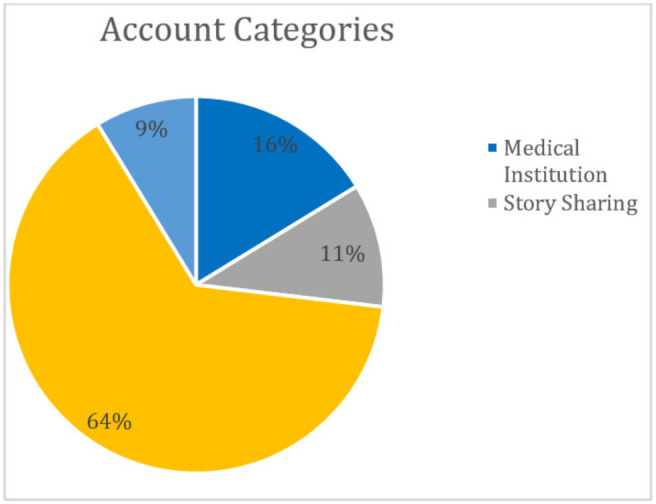
Social media account categories.

**Figure 3 children-08-00869-f003:**
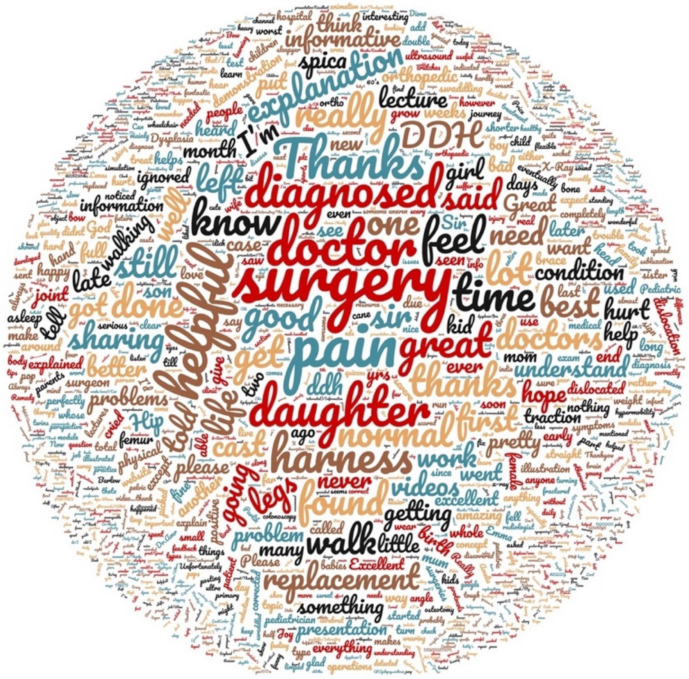
Word cloud of YouTube comments. Larger size indicates more frequently used words.

**Figure 4 children-08-00869-f004:**
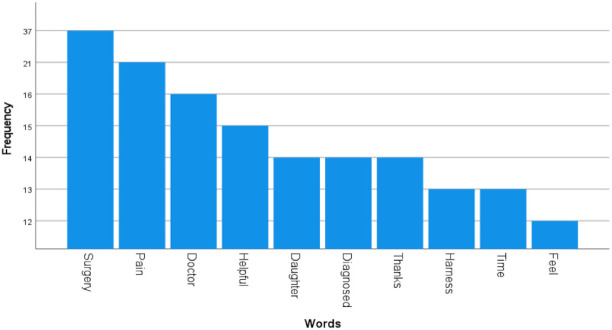
Frequency distribution of the top ten words generated by the word cloud (verbs, adverbs, and common nouns not related to DDH were removed for clarity).

**Table 1 children-08-00869-t001:** Thematic analysis of overall comments.

Themes	Prevalence	Description	Selected Comments
Sharing Information and Advice	36.10%	Experiences and advice regarding everyday care, useful information pertaining to the treatment of children with DDH.	“My daughter is 18 months and has the exact symptoms. We are getting an X-RAY to rule out DDH this week”
Appreciation and Success Stories	9.71%	Sharing success stories of treatment, positive surgical outcomes and thanking providers and persons providing feedback.	“My son is 29 months and just this Monday it was decided to perform traction and in 4 weeks time an Open Reduction. This video helped me to better understand the procedure. Thank you!!!”
Emotional Support	11.35%	Connecting with others based on similar emotional difficulties while taking care of a child with DDH.	“….my baby boy is getting a hip surgery and he’s only 3 yrs old I’m so scared, pray everything comes out fine.”
Social Media as a Second Opinion	21.10%	Offering or receiving recommendations regarding DDH management.	“What types of interventions are necessary after discovering that an infant exhibits either a positive Ortolani’s or Barlow’s sign? Would you take an X-ray? Is surgery necessary for either positive sign?”
Advertisements	6.18%	Advertising for DDH related accessories and equipment.	“Our Pavlik harnesses have been designed to optimally manage DDH. We are giving you the ability to provide a comfortable management of DDH for your child at affordable price”
Challenges and Difficulties	7.08%	Challenges during management including casting, bracing or post-surgery.	“My gosh, so terrible the effects on your little girl from a late diagnosis. Thanks for sharing Joy”
Techniques and Anatomy	8.24%	Sharing technical surgical details regarding DDH corrective surgery and sharing details regarding the anatomy and pathophysiology of DDH.	“Osteotomies should be done when DDH results in more acetabular pathology than femoral pathology (increased acetabular index);”
Inequities and Access Issues	0.25%	Troubles with affording or accessing treatment for DDH.	“Unfortunately, we can’t get the [Brand] here in Canada but it sounds like from other families that the [Brand] usually work well enough”

**Table 2 children-08-00869-t002:** Distribution of comments across, by theme.

Themes	Facebook Page	Facebook Group	YouTube	Twitter
Emotional support and forming connections	1358 (12.3%)	1425 (11.2%)	44 (7.1%)	63 (5.6%)
Sharing information and advice	4947 (45.0%)	3879 (30.5%)	78 (12.5%)	285 (25.5%)
Appreciation and successes	1546 (14.0%)	405 (3.2%)	336 (53.9%)	187 (16.7%)
Challenges and difficulties	573 (5.2%)	1109 (8.7%)	6 (1.0%)	115 (10.3%)
Advertising/offering services	409 (3.7%)	1086 (8.5%)	5 (0.80%)	74 (6.6%)
Inequities and access	31 (0.30%)	31 (0.24%)	9 (1.4%)	4 (0.36%)
Social media as a second opinion	1230 (11.20%)	4006 (31.1%)	28 (4.5%)	103 (9.6)
Information regarding technique and anatomy	909 (8.30%)	786 (6.2%)	117 (18.8%)	287 (25.7%)

**Table 3 children-08-00869-t003:** Mean rank comparison of the number of comment themes amongst social media platforms.

Themes	Facebook Groups	Facebook Pages	YouTube Videos	Twitter	χ^2^	*p* Value
Emotional Support	150.1	146.8	69.2	108.2	89.9	<0.0001 ^1,2,3,4^
Sharing Information and Advice	149.7	146.9	69.2	108.5	84.0	<0.0001 ^1,2,3,4^
Appreciation and Success Stories	142.9	149.5	69.6	115.4	54.6	<0.0001 ^1,2,3,4^
Challenges and Difficulties	151.4	136.7	69.0	112.2	112.4	<0.0001 ^1,2,4^
Advertising Services	152.2	145.2	69.0	112.4	127.0	<0.0001 ^1,2,4^
Inequities and Access Issues	125.2	124.5	72.8	85.2	62.0	<0.0001 ^2,3,4^
Social Media as a Second Opinion	151.8	137.3	69.8	108.6	76.8	<0.0001 ^1,2,3,4^
Techniques and Anatomy	149.5	145.7	69.9	91.8	60.9	<0.0001 ^1,2,3,4^

Pair-wise comparisons were all significant at (*p* < 0.05) except as shown below: ^1^ Twitter vs Facebook pages (*p* > 0.05). ^2^ Twitter vs Facebook groups (*p* > 0.05). ^3^ Twitter vs YouTube (p > 0.05). ^4^ Facebook groups vs Facebook pages (p > 0.05). χ^2^ (Kruskal–Wallis test with Bonferroni correction) was used for multiple comparisons.

## Data Availability

Data available in a publicly accessible repository that does not issue DOIs. Publicly available datasets were analyzed in this study. This data can be found here: https://www.statista.com/markets/424/topic/540/social-media-user-generated-content/#overview.

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
