# Peer review of "Developmental Dysplasia of the Hip: How Does Social Media Influence Patients and Caregivers Seeking Information?"

_children, 2021, doi:10.3390/children8100869_

Round 1

Reviewer 1 Report

It is an original study that has some importance in our actual life style for patients as for caregivers. 

the study is well designed. 

Unofrtunatelly, the results are  quiet difficult to understant, especially table 3. I'm not sure to understand where was a significant difference between whih plat form. Facebook with all others? 

Line 115-122: is not clear. ...that facebook groups and pages contained significantly higher number compared to yotube <0.0001 and then in table 3 you can finde  no difference between facebook groups and pages or between twitter and facebook. Table 3 should be reorganised to become more clear that there was only a significant difference between facebook and youtube. What is the quintescence that the author wants to show?

Author Response

Unofrtunatelly, the results are  quiet difficult to understant, especially table 3. I'm not sure to understand where was a significant difference between whih plat form. Facebook with all others? 

- Table 3 is basically comparing the number of comments in each theme between every platform. For example, for Emotional Support, it is comparing the number of comments in Facebook Groups versus Facebook Pages versus YouTube videos versus Twitter pages and is reporting if there is a statistical significance for each comparison. There was significance in all comparisons with the exception of what is listed at the bottom of the table. 

Line 115-122: is not clear. ...that facebook groups and pages contained significantly higher number compared to yotube <0.0001 and then in table 3 you can finde  no difference between facebook groups and pages or between twitter and facebook. Table 3 should be reorganised to become more clear that there was only a significant difference between facebook and youtube. What is the quintescence that the author wants to show?

- The statement being made is that Facebook Pages and Facebook Groups both have a significantly higher number of comments across all themes compared with YouTube. As you can see the is a significant difference when the Facebook Pages and Groups are compared to YouTube videos in Table 3. If Table 3 seems too confusing or misleading, we are happy to remove it altogether. Please let us know and thank you for your comments. 

Reviewer 2 Report

 Developmental Dysplasia of the Hip: How does social media influence patients and caregivers seeking information?

Reviewer’s comments:

Thank you for this article. There are some points the authors need to address prior to publication

  1. ll. 29: you state that bracing is first line therapy for children with DDH younger than 6 months. This is somewhat misleading as recent literature differentiates between the different severities of DDH. Bracing is used in children with mild DDH (Graf type II), while closed reduction is becoming gold standard in type IV and dislocated hips.

Please adjust your introduction accordingly. You might consider citing this article: Walter SG et al. Closed Reduction as Therapeutic Gold Standard for Treatment of Congenital Hip Dislocation. Z Orthop Unfall. 2020 Oct;158(5):475-480. English, German. doi: 10.1055/a-0979-2346. Epub 2019 Sep 18. PMID: 31533169.

  1. ll. 35: You summarize the importance of social media and its high frequent use among the society. This is fine and the data seem to be evidence based. However, you should try to describe for what purposes social media are used mainly. The widespread availability of social media and certain groups regarding DDH within these platforms does not implicate their impact.

  1. ll. 41: I believe your citations to be somewhat misleading in this context. Without doubt, patients and caregivers do research on social media for specific diseases (actually I believe research is rather done on google than on social media platforms). But I am sure that final advice and treatment decisions etc. is gathered by consulting orthopedic specialists.

I would advice you to somewhat relativize this statement.

  1. Why did you explicitly use Facebook, Twitter and Youtube? Is there a reason for excluding e.g. Google or Instagram? How did make this decision? Please explain in detail.

  1. Figure 1: What does this figure tell the reader? Is DDH becoming of less interest since less groups and videos have been created since 2015? Or are information available already sufficient? Or have parents and caregivers moved forward to other platforms to gain information?

  1. Figure 2: please remove the category “news” if it presents 0%

  1. Table 2: from my point of view this table can be summarized as: “social media on DDH largely provide secondary and emotional information; medical information are found only to a minor percentage”

Please discuss this point in the discussion section ( I acknowledge you critical remarks in ll. 199 -201).

Author Response

Thank you for your review and comments. Please find our responses below. 

  1. ll. 29: you state that bracing is first line therapy for children with DDH younger than 6 months. This is somewhat misleading as recent literature differentiates between the different severities of DDH. Bracing is used in children with mild DDH (Graf type II), while closed reduction is becoming gold standard in type IV and dislocated hips.

Please adjust your introduction accordingly. You might consider citing this article: Walter SG et al. Closed Reduction as Therapeutic Gold Standard for Treatment of Congenital Hip Dislocation. Z Orthop Unfall. 2020 Oct;158(5):475-480. English, German. doi: 10.1055/a-0979-2346. Epub 2019 Sep 18. PMID: 31533169.

- Thank you for the suggestion. I have edited the introduction and cited the article you provided. 

  1. ll. 35: You summarize the importance of social media and its high frequent use among the society. This is fine and the data seem to be evidence based. However, you should try to describe for what purposes social media are used mainly. The widespread availability of social media and certain groups regarding DDH within these platforms does not implicate their impact.

- We have made some edits in the following paragraph to address this. 

  1. ll. 41: I believe your citations to be somewhat misleading in this context. Without doubt, patients and caregivers do research on social media for specific diseases (actually I believe research is rather done on google than on social media platforms). But I am sure that final advice and treatment decisions etc. is gathered by consulting orthopedic specialists.

I would advice you to somewhat relativize this statement.

- We have edited this sentence to be less misleading. Please let us know your thoughts.

  1. Why did you explicitly use Facebook, Twitter and Youtube? Is there a reason for excluding e.g. Google or Instagram? How did make this decision? Please explain in detail.

- Google is not considered a social media platform therefore we did not use it in our study. Instagram was very difficult to extract data from and there were not many accounts to begin with, therefore the decision was made to exclude it from this study. I added this into our materials/methods as well, please let us know if you would like us to keep the explanation in the manuscript. 

  1. Figure 1: What does this figure tell the reader? Is DDH becoming of less interest since less groups and videos have been created since 2015? Or are information available already sufficient? Or have parents and caregivers moved forward to other platforms to gain information?

- This figure is simply showing the trend of account creation over the years, and the decrease over recent years may prompt orthopaedists to create more up to date and accurate content. 

  1. Figure 2: please remove the category “news” if it presents 0%

- We have removed this. 

  1. Table 2: from my point of view this table can be summarized as: “social media on DDH largely provide secondary and emotional information; medical information are found only to a minor percentage”

Please discuss this point in the discussion section ( I acknowledge you critical remarks in ll. 199 -201).

- We have included this point in the discussion. Thank you very much for your suggestions.

Round 2

Reviewer 2 Report

Thank you for this article. I will help open a new discussion among caregivers on how to best provide medical information to patients on recent social media platforms